# Role of Extracellular Vesicles in the Pathophysiology, Diagnosis and Tracking of Non-Alcoholic Fatty Liver Disease

**DOI:** 10.3390/jcm9072032

**Published:** 2020-06-29

**Authors:** Lauren A. Newman, Michael J. Sorich, Andrew Rowland

**Affiliations:** College of Medicine and Public Health, Flinders University, Adelaide, SA 5042, Australia; michael.sorich@flinders.edu.au (M.J.S.); andrew.rowland@flinders.edu.au (A.R.)

**Keywords:** non-alcoholic fatty liver disease, non-alcoholic steatohepatitis, extracellular vesicles, biomarkers

## Abstract

Non-alcoholic fatty liver disease (NAFLD) is the most common chronic liver disease, affecting approximately one-third of the global population. Most affected individuals experience only simple steatosis—an accumulation of fat in the liver—but a proportion of these patients will progress to the more severe form of the disease, non-alcoholic steatohepatitis (NASH), which enhances the risk of cirrhosis and hepatocellular carcinoma. Diagnostic approaches to NAFLD are currently limited in accuracy and efficiency; and liver biopsy remains the only reliable way to confirm NASH. This technique, however, is highly invasive and poses risks to patients. Hence, there is an increasing demand for improved minimally invasive diagnostic tools for screening at-risk individuals and identifying patients with more severe disease as well as those likely to progress to such stages. Recently, extracellular vesicles (EVs)—small membrane-bound particles released by virtually all cell types into circulation—have emerged as a rich potential source of biomarkers that can reflect liver function and pathological processes in NAFLD. Of particular interest to the diagnosis and tracking of NAFLD is the potential to extract microRNAs miR-122 and miR-192 from EVs circulating in blood, particularly when using an isolation technique that selectively captures hepatocyte-derived EVs.

## 1. Introduction

Estimates of the global prevalence of non-alcoholic fatty liver disease (NAFLD) range from 25 to 35% [1]. To provide context, this equates to more than 1 billion individuals living with this disease. The majority of individuals with NAFLD do not experience symptoms, but up to one in three will progress to the more severe form of the disease, non-alcoholic steatohepatitis (NASH), which is a major risk factor for life-threatening cirrhosis or hepatocellular carcinoma (HCC). Given the asymptomatic nature of the early stages of disease, reliable non-invasive mechanisms to identify affected individuals—in particular, those at greatest risk of progressing to more severe forms of the disease—are urgently needed. Consistent with the major burden of disease associated with NAFLD, considerable resources are being dedicated within the pharmaceutical industry and academia to develop novel treatments for this disease, with estimates that up to 40% of new molecular entities (NMEs) currently in development target NAFLD. In terms of drug development, the absence of a robust and dynamic approach to track disease progression and treatment response hinders the efficiency of development programs and prevents resources from being targeted to the most promising NME candidates.

The early stage of NAFLD is characterised by accumulation of fat in the liver, referred to as simple steatosis (SS) or non-alcoholic fatty liver (NAFL) [2]. By definition, NAFLD can only be diagnosed in patients with no history of significant alcohol consumption, which is specified by the US guidelines for NAFLD as alcohol intake less than 21 standard drinks per week for males and less than 14 standard drinks per week for females [3]. Other causes of hepatic steatosis, such as viral hepatitis and medications, must also be excluded [4]. Risk factors for NAFLD include obesity, insulin resistance with or without type 2 diabetes, dyslipidaemia and other metabolic disturbances, and hence the disease is generally considered a hepatic manifestation of metabolic syndrome [5]. However, aetiology is not exclusively diet related, as NAFLD can occur in non-obese individuals and has been associated with ethnicity and genetic polymorphisms [6].

In parallel to the emergence of NAFLD as a major health burden, the role of extracellular vesicles (EVs) in cellular communication, particularly during disease, has gained significant recent attention. EVs are a heterogenous family of small membrane-encapsulated particles constitutively released by virtually all cell types into various biological fluids such as blood, saliva and urine [7,8,9]. While individual EV types (e.g., exosomes and microvesicles) differ in terms of their origin and composition, they all contain an array of molecular cargo including microRNA (miRNA), messenger RNA (mRNA) and protein that is derived from their parent cell. The potential to harvest this cargo and utilize it as a source of diagnostic information in NAFLD represents an intriguing potential to address the need for a minimally invasive, dynamic approach to track the burden of disease.

In this review, we will summarise key information regarding the prevalence and pathogenesis of NAFLD, with a focus on the role of EVs in disease development and progression. The performance of current approaches to diagnose and track NAFLD will then be considered and insights regarding the most promising EV-derived biomarker strategies will be highlighted.

### Prevalence and Significance of Health Burden

NAFLD is the most common chronic liver disease in the world in both adult and paediatric populations and its prevalence continues to climb with the rate of obesity and other metabolic disturbances [6,10]. The prevalence of NAFLD has been reported to be as high as 38% of the general population, but this estimate varies across studies utilizing different diagnostic techniques [1]. Of the individuals affected by NAFLD, up to 30% go on to develop NASH [11], which extrapolates to approximately 3–15% of the general population [9]. In addition to the direct disease burden, the liver serves as the major clearance organ for >70% of small molecule drugs, with drugs from essentially all therapeutic classes being cleared by cytochrome P450 (CYP) and UDP-glucuronosyltransferase (UGT) enzymes [12,13]. Pathological changes caused by NAFLD are known to alter the expression and metabolic activity of CYP and UGT proteins [14], and represent an emerging important source of variability in drug exposure that is important to account for in drug development and the clinical use of medicines [15].

## 2. Clinical Spectrum of Non-Alcoholic Fatty Liver Disease

Clinically, NAFLD presents as a broad spectrum of disease (Figure 1). Most affected individuals only exhibit simple steatosis (SS), which is relatively benign, often asymptomatic and appears to not significantly reduce life expectancy [2,5]. Approximately three out of every ten patients progress to the more severe form of the disease (NASH), which manifests as hepatocyte injury and cell death, hepatic inflammation with marked immune cell infiltration and varying degree of fibrosis [16]. Liver damage in NASH has the potential to further progress to cirrhosis and end-stage liver failure or hepatocellular carcinoma (HCC) [17]. The highly heterogenous nature of the disease is a result of variable and multifaceted aetiopathogenesis. SS does not inevitably lead to NASH or related progressive liver damage, rather variability between and within individuals at any given time governs disease development. More specifically, there are alterations to the mechanisms involved in lipid flux in hepatocytes, including de novo lipogenesis (DNL), uptake from systemic and portal circulations and export of very low-density lipoproteins (VLDL), the composition of lipid moieties with varying cytotoxic potential that accumulate in hepatocytes and individual ability to defend against lipotoxicity via metabolic adaption [11,18]. Differences in individual inflammatory and wound-healing responses also cause heterogeneity in NASH severity [11,18].

## 3. Pathogenesis of NAFLD and NASH

The pathogenesis of NAFLD, including mechanisms for the development of hepatic steatosis and progression to NASH, are complex, multifaceted and yet to be fully elucidated [5,9]. Over the past two decades, different theories have been proposed that attempt to explain the development and broad clinical spectrum of NAFLD [9]. According to the initial theory, termed the “two hits” hypothesis, metabolic factors leading to hepatic accumulation of triacylglycerides (TAGs), such as a high-fat diet, obesity and insulin resistance, serve as the “first hit”, which sensitises hepatocytes to the “second hit” and promotes progression to NASH. This “second hit” comprises signalling via cytokines and adipokines, mitochondrial and ER stress, inflammatory processes, and fibrogenesis [9]. While the importance of each element remains apparent, the traditional understanding of NAFLD pathogenesis has evolved into a new theory known as the “multiple parallel hits” hypothesis, which accounts for the complex interplay of said elements and recognises that they may not act in a distinct sequence [5].

### 3.1. Development of Hepatic Steatosis

Hepatic steatosis occurs when the influx or synthesis of lipids exceeds their efflux or degradation [18]. Fatty acids are derived from the diet, DNL and adipose tissue via lipolysis and, in hepatocytes, are converted to TAG for storage in lipid droplets, exported as VLDL or oxidised in mitochondria or peroxisomes [9,19]. Adipose tissue plays a key role in lipid homeostasis, not only as a primary location of TAG storage but also via endocrine activity. Adipose tissue dysfunction can occur with metabolic disturbances, particularly obesity, wherein chronic energy surplus and excess visceral adiposity increases the free fatty acid (FFA) source delivered to the liver [9]. Additionally, cytokines such as tumour necrosis factor-α (TNF-α) and adipokines (e.g., resistin) are produced and act to reduce the sensitivity of adipocytes to hormones, particularly insulin, that normally promote the storage of fat. Consequently, fatty acids are less readily integrated in adipocytes and their release from fat deposits is enhanced [18]. Exposure of hepatocytes to the increased concentration of circulating FFAs leads to increased uptake, which is further promoted since hepatocytes express receptors for adipokines and adipogenic hormones [11,18]. Hepatocytes attempt to adapt by upregulating TAG formation or oxidation in mitochondria and peroxisomes. Despite the efficiency of these processes, the cells may not be able to meet the persistently high metabolic demand [11], and moreover, defects in insulin signalling can result in both an increase in DNL and inhibition of β-oxidation—all of which promote lipid accumulation in hepatocytes [9].

### 3.2. Lipotoxicity and Lipoapoptosis

Triglycerides are not inherently hepatotoxic [6,18]. Rather, cell injury can occur in the presence of other lipid moieties, such as FFA, free cholesterol and intermediate metabolic products, including diacylglycerol and phospholipids (e.g., sphingolipids and ceramides) [2,6,17,18]. Hepatic lipotoxicity is cellular stress resulting from the accumulation of these toxic lipids that cannot be accommodated by the metabolic capacity of the cell [11]. The generation of reactive oxygen species (ROS) is a detrimental result of increased hepatic fatty acid oxidation, and when the rate of this remains high, anti-oxidant stores are eventually depleted, thus increasing the vulnerability of hepatocytes to oxidative stress [18,19]. Lipotoxicity also elicits endoplasmic reticulum (ER) stress and initiates stress–response pathways, such as the c-Jun N-terminal kinase (JNK) pathway, which upregulates proapoptotic proteins and death receptor expression (Figure 2) [17].

A defining feature of NASH is when lipotoxicity becomes lethal to hepatocytes, which occurs predominantly via a process known as lipoapoptosis [11,16]. Death receptors (DR) belong to the superfamily of TNF receptors and when activated on the cell surface, initiate the extrinsic pathway of apoptosis. The liver primarily expresses Fas, TNF receptor 1 and TNF-related apoptosis-inducing ligand (TRAIL) receptor 1 and 2. Upregulation of these receptors has been demonstrated in mouse models of NAFLD and human NASH liver samples [17]. TRAIL-R2, also known as DR5, is arguably the most important death receptor for apoptotic signalling in lipotoxic hepatocytes [16,17]. Ligand binding to DR5 triggers oligomerisation of the extracellular domain which initiates intracellular signalling resulting in caspase-dependent cell death. However, in lipotoxic hepatocytes, the saturated fatty acid palmitate and other toxic metabolites, especially lysophosphatidylcholine (LPC) and ceramide, not only upregulate the expression of pro-apoptotic proteins but cause rearrangement of DR domains that permits ligand-independent activation of DR5 (Figure 2) [16,17].

### 3.3. Inflammation

In NASH, damaged and dying lipotoxic hepatocytes express damage-associated molecular patterns (DAMPs) that signal their impending death to neighbouring cells and initiate wound-healing responses [2,11]. These are recognised by toll-like receptors (TLRs) of the innate immune system and lead to the activation of liver-resident macrophages, known as Kupffer cells, expression of chemokines and pro-inflammatory cytokines, such as TNF-α and interleukin 6 (IL-6), and large-scale recruitment of circulating monocytes, neutrophils and natural killer (NK) cells to the liver [2,9]. In particular, TLR4 is widely expressed across liver cell types and can be activated directly by FFAs [2,17]. Moreover, FFA-mediated activation of TLR4 on adipocytes, as well as cytokine signalling, exacerbates peripheral insulin resistance [17]. Essentially, lipotoxic cell death in NASH drives “sterile” inflammation—i.e., inflammation that occurs in the absence of pathogens—in a feed-forward loop, as sustained inflammation exacerbates hepatocyte injury and inflammatory cells express ligands for Fas and TRAIL receptors that contribute to cell death [2,16].

### 3.4. Fibrogenesis

Normally, DAMP expression and resultant inflammation is necessary for repair, and is downregulated when the regenerative process approaches completion. In NASH, however, the activity of inflammatory and other repair-related cells, such as myofibroblasts and liver progenitors, persists due to chronic lipotoxicity [11]. Hepatic stellate cells (HSCs) are liver pericytes that produce components of the extracellular matrix (ECM), such as collagen and glycoproteins, when activated [11]. This switch to a myofibroblast-like phenotype occurs in response to cytokines and profibrogenic factors released by activated Kupffer cells, other infiltrating inflammatory cells and injured hepatocytes, or when quiescent HSCs take up apoptotic bodies or other extracellular vesicles released from hepatocytes [11,16]. Fibrogenesis is also promoted by ballooned hepatocytes. Ballooning is cellular degradation characterised by swelling, a centralised nucleus and loss of cytoskeletal integrity and is a sublethal result of lipotoxicity, in which cell death is initiated but unable to be completed [2]. This state induces chronic secretion of various factors that promote tissue remodelling and contribute to fibrosis in NASH, especially hedgehog ligands [17]. Fibrosis often increases in severity over time and may advance to hepatic cirrhosis. Patients with this pathology are at greater risk of developing HCC. However, the activation of signalling pathways with implications in carcinogenesis suggests that NAFLD may also promote HCC in stages preceding cirrhosis [6].

## 4. Methods for Diagnosing and Staging NAFLD

NAFLD diagnosis is made when fatty change is observed by imaging or histology in more than 5% of hepatocytes or of total liver volume [10]. Other causes of steatosis or chronic liver diseases must also be ruled out, and this includes a history of significant alcohol consumption [3].

### 4.1. Liver Biopsy

Evaluation of a liver biopsy is the gold standard for NAFLD diagnosis and the only way to accurately differentiate between simple steatosis and NASH [1]. Histological hallmarks include mixed macrovesicular and microvesicular steatosis, hepatocyte ballooning degeneration, lobular inflammation and fibrosis [1,10]. These features are assigned a numerical value to give the NAFLD activity score (NAS), a system developed by the NASH Clinical Research Network to grade NAFLD across its clinical spectrum [1,3,20]. However, there are several limitations to this method of diagnosis. Liver biopsy is an invasive approach that comes with risks such as pain, bleeding, organ perforation and even death, and consequently is selectively indicated for NAFLD patients with a greater likelihood of progression and rarely used repeatedly [10,21].

### 4.2. Abdominal Imaging and Elastography

Abdominal imaging is sometimes sufficient to identify patients with steatosis without the need for liver biopsy [10]. The first-line approach is ultrasound, which is simple and inexpensive. However, this technique has low sensitivity in patients with less than 30% steatosis [1,3]. Computed tomography (CT) and magnetic resonance imaging (MRI) provide more precise quantification of fatty change but are more expensive and still cannot accurately diagnose NASH [1,10]. Transient elastography is another non-invasive approach that measures hepatic stiffness, thus providing an estimate of fibrosis. This is a quick and easy test but is limited by a lack of precise cutoff for different stages of fibrosis and technical issues arise in patients with excess abdominal fat, as it attenuates the transmitted vibrations [22].

### 4.3. Alternative Non-Invasive Diagnostic Tests

Various parameters, including serum levels of the liver enzymes alanine and aspartate aminotransferases (ALT, AST) and gamma glutamyl transferase (GGT), triglycerides, BMI, waist circumferences, presence of diabetes, fasting insulin and results of ultrasonography or magnetic resonance spectroscopy (MRS) may be accounted for in predictive scores or indices for NAFLD screening. Different combinations of these parameters produce, for example, the fatty liver index, the hepatic steatosis index and the liver fat score. Studies have demonstrated strong predictive performance for scores and indices based on analysis of area under the receiver operator curve (AUROC) values of up to 0.87. However, limitations in precision and practicality remain, such as the lack of routine tests for some parameters and the use of ultrasonography (sub-optimal sensitivity) as a reference standard [23]. Further, liver enzymes have exhibited poor predictive value (specificity), as they may be elevated in individuals without disease or due to other pathologies [24]. Importantly, the results of these minimally invasive tests have typically demonstrated insufficient reproducibility between studies to warrant use in routine care [25]. No test has yet been able to replace liver biopsy or determine, at earlier stages, which patients will join the subset that progress to more severe disease [18,20]. Thus, there is a pressing need for an accurate and minimally invasive strategy to diagnose NAFLD across its full clinical spectrum [21]. A serum biomarker of notable performance is cytokeratin-18 (CK-18), which is an intermediate filament protein fragment released during hepatocyte apoptosis or necrosis and, therefore, increased in NASH serum compared to SS and healthy individuals [23]. CK-18 has promising diagnostic value with AUROC for NASH diagnosis reported as high as 0.81 [26]. However, performance is inconsistent and most studies conclude insufficient sensitivity and specificity to differentiate stages of disease [25,27]. More broadly, elevated CK-18 has been demonstrated to correlate with increasing severity of liver dysfunction [28]. As such, it is likely that while useful in defining the severity of injury, this marker is not well suited to discriminate NAFLD from other types of liver injury.

## 5. Extracellular Vesicles

Extracellular vesicles (EVs) are small membrane-bound particles constitutively released by virtually all cell types and can be detected in a range of biological fluids, including blood, saliva, urine, cerebrospinal fluid, breast milk, and amniotic fluid [7,8,9]. EVs are important mediators of intercellular communication within tissues and between organs upon entry into the systemic circulation [29]. This activity is important in the maintenance of homeostasis but is also implicated in the pathogenesis of major human diseases including metabolic syndrome [30,31]. EVs contain a diverse molecular cargo, consisting of proteins, lipids and nucleic acids, including mRNA, miRNA, other non-coding RNAs, and DNA. Circulating EVs may undergo membrane fusion and subsequent internalisation by target cells or trigger signalling pathways via receptor–ligand interactions on the surface [7,8]. Horizontal transfer of molecular cargo results in phenotypic and functional changes in recipient cells’ activity, such as enhancing or suppressing specific gene expression and promoting cell proliferation or differentiation [8,32]. Classification of EV subtypes is based on their subcellular origin and mechanism of biogenesis. The two main populations of EVs, referred to collectively as small EVs (sEVs), include exosomes, which are formed by internal budding of endosomal membranes and range from 30 to 150 nm in diameter; and microvesicles (MVs), sized at 50–1000 nm, that bud directly from the cell membrane [7,33]. Apoptotic bodies and oncosomes are larger populations of EVs that are released during apoptotic cell death and by migratory cancer cells, respectively, but are beyond the scope of this review [8].

### 5.1. Biogenesis and Release of Small EVs

Exosomes are formed in the process of endosomal maturation, whereby the early endosome matures into a multivesicular body (MVB) and this membrane undergoes invagination to form intraluminal vesicles (ILVs) (Figure 3). MVBs are transported to and fuse with the plasma membrane, subsequently releasing their luminal contents into the extracellular environment. At this point, the ILVs are considered exosomes [34,35]. Some MVBs, however, are directed to lysosomal degradation where the contents are recycled. The regulation of this balance is not well understood but is likely influenced by the activity of small Rab guanosine triphosphatases (GTPases), due to their critical involvement in trafficking of MVBs to the plasma membrane and promotion of membrane fusion and exosome release [7,36]. Rab35, for example, is associated with an increase in ILVs at the cell surface [37] and silencing of Rab27 isoforms was shown to decrease exosome release from cells in culture [38]. Pathways of exosome biogenesis are traditionally considered dependent or independent of endosomal sorting complex required for transport (ESCRT) [34]. ESCRT-dependent biogenesis involves the formation of ESCRT complexes 0, I, II, III, which mediate invagination and scission of endosomal membranes. However, in ESCRT-independent biogenesis, ceramide is incorporated into lipid rafts in a process highly reliant on neutral sphingomyelinase 2 (nSmase2), which promotes inward budding of endosomal membranes [34,39]. Though the existence of more than one distinct pathway for biogenesis may account for some of the heterogeneity observed in exosome populations, it is suggested that the pathways work synergistically and may vary in their relative contributions to exosome production, dependent on the cell type or in response to different stimuli [33,34]. In comparison to exosomes, considerably less is known about the molecular mechanisms underlying MV release [7]. MVs are not of endosomal origin, rather production occurs by outward budding of the cell membrane. At the site of MV shedding, lipid domains, such as phosphatidylserine, are redistributed in the plasma membrane, the local cytoskeletal network is reorganised and the actin–myosin machinery contracts to alter membrane curvature and promote budding and detachment of vesicles. Notably, ADP-ribosylation factor 6 (ARF6), ESCRT components and activity of RhoA with its effector kinase ROCK1, mediated via activated caspase 3, are involved in the regulation of this process. Intracellular calcium levels also influence MV release (Figure 3) [7,33].

### 5.2. Molecular Cargo of Small EVs

Encapsulation of molecular cargo in circulating EVs grants protection from degradation, thus facilitating efficient transfer of intended signals between cells [30]. Cargo reflects the cytosolic composition and physiological state (i.e., quiescent, stimulated, stressed or transformed) of the parent cell, in addition to the mechanism by which the sEV was produced [32]. Common exosome markers include tetraspanins/cluster of differentiation proteins, such as CD9, CD81 and CD63, heat shock proteins, Hsp70 and Hsp90, ESCRT components and other proteins involved in the endosomal pathway, such as tumour susceptibility gene 101 (TSG101) and ALG-2-interacting protein X (ALIX) [29,35], while Annexin A1 was recently identified as a microvesicle-specific marker [40]. sEVs also contain an assortment of proteins, lipids and RNAs, especially miRNA, that may reflect levels of these species in the parent cell. However, proteins that exist in high cytosolic abundance, such as cytoskeletal or glycolytic proteins, are relatively lacking in EVs which points to a more regulated process for loading molecular cargo [40]. Diversity has been observed in EV pools from different cell types as well as from the same cell. Though the current understanding of the mechanisms underlying this is limited, studies have pointed to modes of selective packaging within the different pathways of sEV biogenesis [33]. Furthermore, cell stress and other stimuli often alter the nature of the signals that need to be conveyed between neighbouring cells and so have a profound effect on sEV biogenesis and contents [41].

### 5.3. Selective Packaging of RNA and Other Molecular Cargo

Sorting of molecular cargo into sEVs is reported to occur in an active and selective manner [40]. Research efforts focus on understanding the interactions that promote the selection and localisation of RNA species into EVs, but this work has mainly elucidated packaging into exosomes while knowledge of the molecular mechanisms for MVs remain limited [33]. Loading of exosomal RNA appears to occur during early stages of MVB development as RNA molecules are recruited to lipid raft-like regions mediated by hydrophobic modifications and membrane reorganisation [33]. miRNAs are important exosomal cargo, as these small non-coding RNAs of 17–24 nucleotides make up the majority of exosomal nucleic acid content and are key regulators of gene expression [30,42]. The expression of more than 60% of human mRNA may be controlled by the targeted binding of miRNA to 3′ untranslated regions (UTR) or open reading frames (ORF), resulting in post-transcriptional suppression and diverse changes in cellular function [30].

Profiling studies have demonstrated non-random sorting of miRNA into EVs—for example, a subset of which are preferentially incorporated in exosomes secreted from a range of cell types, with miR-451 most consistently detected [43]. Some mechanisms of miRNA loading are beginning to be defined and are likely to be influenced by the pathway of EV biogenesis. In the ESCRT-independent pathway, miRNA content was shown to increase and decrease in response to nSMase2 overexpression and inhibition, respectively [44]. ESCRT machinery has also been implicated with miRNA induced silencing complex formation (miRISC) and localisation of miRNAs to MVBs [30,45]. Post-transcriptional uridylation or adenylation of 3′ ends on small non-coding RNAs may also provide a signal for selective loading as enrichment of these modified species have been observed in cells and exosomes, respectively [30,33]. Similarly, variants of a particular 25 nucleotide sequence in the 3′UTR of mRNA was observed to be enriched in MVs [33]. Furthermore, specific nucleotide motifs on miRNAs are shown to be recognised and bound by heterogenous nuclear ribonucleoproteins (hnRNPs) to control sorting into vesicles. Of importance is sumoylated hnRNPA1B2 which recognises a GGAG motif in miRNA 3′ ends [30,33] and, while two other members of this protein family, hnRNPA1 and hnRNPC, are known to bind exosomal miRNAs, their specific binding motifs are yet to be determined [30].

Lastly, synaptotagmin-binding cytoplasmic RNA-interacting protein (SYNCRIP) also has a known function in miRNA sorting. Importantly, not all EV-associated RNA is encapsulated and this is demonstrated by an approximately 7% reduction in RNA yield when EV pellets are treated with RNase. Hence, the sorting mechanisms distinguishing RNA species that are packaged or those bound to the outer surface of EVs, complexed with RNA-binding or chaperone proteins, remain to be investigated further [33]. Like RNA, selective protein packaging in exosomes and MVs may be affected by components involved in EV biogenesis pathways, particularly ESCRT. Silencing of ALIX, a protein that binds to ESCRT-3, was shown to alter protein composition while exosome secretion remained unaffected [34]. Ubiquitination of proteins is known to promote their lysosomal degradation but may also be a mechanism for targeting to ILVs via ESCRT-0 [8,34]. Further, tetraspanins, including Tspan8, are involved in ILV formation and may alter protein and mRNA content, and Need family interacting protein 4 (NEDD4) has been implicated in targeting cytosolic proteins to ILVs. Sorting proteins into MVs is facilitated by several factors, including ARF6, Rab22a and vesicle-associated membrane protein 3 (VAMP3) which deliver specific cargo to sections of high membrane budding [33].

### 5.4. Effect of Physiology Condition of Parent Cell on sEV Release

Secreted EVs are reflective of the parent cell’s functional state and, while released continuously under normal physiological conditions, stressful or pathogenic stimuli may increase abundance or modify the cargo [17]. This may be a means of communicating stress to neighbouring cells or enhancing the elimination of unwanted products by EVs targeted to phagocytes [34]. Several studies have investigated the effect of different culture conditions on cellular EV release and demonstrated alterations in protein and RNA composition as a result of hypoxia, nutrient-deficiency, extracellular pH changes, or stress-inducing ligands, such as lipopolysaccharide or hydrogen peroxide [33,46]. Moreover, Kanemoto [41] reported the induction of MVB formation and increase in exosome release via ER stress responses mediated by ER stress sensors inositol-requiring enzyme 1 (IRE1) and protein kinase RNA-like ER kinase (PERK).

## 6. Role of Small EVs in NAFLD

The liver is a large complex organ consisting of a range of cell types with diverse functions in metabolism, detoxifying blood, and the synthesis, processing and storage of important nutrients [29]. These are mostly performed by the parenchymal cells, or hepatocytes, which occupy the majority of total liver volume. The non-parenchymal cells, including HSCs, liver sinusoidal endothelial cells (LSECs), Kupffer cells and cholangiocytes, are also numerous and responsible for regulation and support of hepatocyte activity [8,29,35]. Effective communication between these different cell types is crucial for proper physiological function and maintenance of homeostasis and it is now well-established that EVs participate in this process and are ubiquitously released in the liver [7]. However, the activity of EVs also extends to the initiation and progression of disease, as pathophysiological conditions alter their abundance and the molecular signals delivered to neighbouring cells [8,9]. Both exosomes and MVs have demonstrated contributions to several facets of NAFLD’s complex pathogenesis, particularly in promoting hepatic cell death, inflammation, pathological angiogenesis and fibrogenesis. Recent work in this [15,47] and other [48] laboratories has further demonstrated the capacity to quantify the expression and activity of CYP and UGT protein and mRNA in plasma derived EV. These data raise the intriguing potential to directly track the impact of NAFLD associated changes on CYP and UGT mediated drug clearance using EV-derived biomarkers, and account for variability in exposure, which can impact drug efficacy and tolerability [49,50,51]. These drug-metabolising enzymes represent a major class of liver enriched markers in circulating EVs and, like CK-18 are known to exhibit marked changes in expression in disease. As such, along with other liver-specific markers such as vanin-1 and asiaglycoprotein receptors 1 and 2, quantification of these markers in EVs may have a role in the grading the severity of liver injury. Notably, as with CK-18, CYP and UGT expression is also altered by a number of other forms of liver disease, making it unlikely that these markers will be able to distinguish NAFLD from other forms of liver pathology.

### 6.1. Hepatocyte-Derived sEVs in NAFLD

Mouse models and human subjects with NASH demonstrate increased levels of circulating EVs and it is suggested that this pool is largely contributed to by hepatocytes [29]. Lipotoxic conditions are known to enhance EV release from hepatocytes in vitro [52] and one study by Povero [53] demonstrated an increase in liver-abundant protein and miRNA cargo in circulating EVs from an experimental NASH model. In this model, mice fed a choline-deficient L-amino acid (CDAA) diet had progressively increased levels of circulating EVs which correlated with the development of histological features of NASH. Protein cargo was reflective of disease processes and could differentiate CDAA-fed and control mice. Mitochondrial DNA, which is implicated with TLR-9 activation and characteristic sterile inflammation, was also increased in hepatocyte-derived EVs from mice and humans with NASH [54]. Several other studies have reported the contribution of EVs to the feed-forward loop of hepatocyte lipotoxic injury and hepatic inflammation. Hirsova [55] exposed hepatocytes to palmitate or LPC to induce lipotoxicity in vitro which resulted in 3-fold greater EV release and increased mean size.

These vesicles contained TRAIL and, when delivered to mouse bone marrow-derived macrophages, induced expression of interleukins IL-1β and IL-6 mRNA. TRAIL and other DR-ligand expression on EVs and activated macrophages also propagates hepatocyte cell death [16]. This study determined that MVs comprised the majority of lipotoxicity-induced EVs released via DR5 proapoptotic signalling, since genetic or pharmacological approaches to ROCK1 inhibition significantly reduced EV release, while Rab27 knockout had no effect [55]. Palmitate treatment has also been shown to activate all three ER stress sensors, IRE1α, PERK and activating transcription factor 6α (ATF6α) and drive ceramide synthesis at the ER. Ceramides are bioactive lipids that play a role in exosome biogenesis and have been implicated with insulin resistance and NASH. EVs isolated from palmitate-treated hepatocytes were not only found to be enriched in ceramides in an IRE1α-dependent process, but also chemoattractive towards macrophages in vitro (Figure 4). This observation was attributed to EV carriage of sphingosine-1-phosphate (S1P), which is formed by phosphorylation of ceramide-derived sphingosine via sphingosine kinases (SphK)-1 and -2. Macrophage activation and chemotaxis was significantly reduced by SphK inhibition [56]. Ibrahim [57] reported that lipotoxic hepatocyte-derived EVs, positive for both exosome and MV markers, also expressed C-X-C motif ligand 10 (CXCL10), a powerful macrophage chemoattractant. Their release was implicated with mixed lineage kinase 3 (MLK3), the mitogen-activated protein kinase that mediates lipotoxicity-induced JNK activation in the liver. JNK inhibition reduced EV release by lipotoxic hepatocytes and impaired CXCL10 trafficking during EV formation. This extends from previous work by the same group whereby MLK3 knockout mice exhibited decreased circulating EVs and CXCL10 content and were protected against diet-induced steatohepatitis [58].

Hepatocyte-derived sEVs also have a well-documented role in promoting liver fibrosis in NAFLD, via direct modulation of HSC phenotype. Povero [59] observed efficient internalisation of lipid-induced hepatocyte-EVs into HSCs and subsequent upregulation of profibrogenic genes, including collagen-1α1 (Col1α1), α-smooth muscle actin (αSMA), and tissue inhibitor of metalloproteinase-2 (TIMP-2), and cell proliferation and migration. Peroxisome proliferator-activated receptor-gamma (PPARγ) is crucial in the maintenance of HSC quiescent phenotype in normal liver and its expression is eventually depleted in the process of HSC activation. PPARγ is targeted by miR-128-3p which, in this study, was enriched in EVs from palmitate-treated hepatocytes and its inhibition abrogated the phenotypic switching of HSCs. These EVs were also enriched in Vanin-1 (VNN1), an ectoenzyme involved in cell adherence and migration. Application of a neutralising antibody against VNN1 reduced internalisation of EVs and the effects on HSCs, suggesting a vital role in the shuttling of miR-128-3p in NAFLD. This author had previously reported that VNN1 was upregulated on MVs released from lipotoxic hepatocytes and mediated their internalisation by LSECs. This resulted in endothelial cell migration and tube formation in vitro and pathological angiogenesis in a NASH mouse model (Figure 4) [60]. Lee [52] observed an alteration of miRNA cargo in exosomes from palmitic acid-treated hepatocytes, particularly a 5-fold increase in miR-122 and miR-192 expression compared to control cells. This study also reported a dose-dependent increase in expression of fibrosis-associated genes when the EVs were applied to HSCs.

Exosomal miR-192 may account for some of this profibrogenic activation, as its transfection into HSCs also increased expression of αSMA, Col1α1 and transforming growth factor-β (TGF-β). Another study suggested that altered miR-122 expression results in fibrosis as its decrease in liver tissue correlated with an increase within circulating EVs and marked liver fibrosis in a mouse model of NASH. The authors reported that novel miR-122 target genes, hypoxia-inducible factor-1α (HIF-1α), vimentin and MAP3K are increased in murine NASH liver and contribute to tissue remodelling [61].

### 6.2. Extra-Hepatic and Hepatic Non-Parenchymal Cell-Derived sEV in NAFLD

Non-parenchymal cells comprise another source of sEVs that have demonstrated contributions to NAFLD pathogenesis and progression. For example, patients with NAFLD had increased circulating MVs derived from macrophages and NK cells which correlated with histological severity, which underlies the increased innate immune activity during development of steatohepatitis [62]. Endothelial cell-derived exosomes contribute to HSC activation and migration via transfer of SphK1 and its substrate S1P [63]. Activated HSCs were shown to communicate phenotypic switching to other HSCs via EVs containing connective tissue growth factor (CTGF) or miR-214, thereby proliferating profibrogenic activation throughout the tissue [64]. Moreover, EVs from cholangiocytes and activated HSCs contain hedgehog ligands that induce proangiogenic activation in LSECs and neovascularisation in rats [65]. Interorgan cross-talk, particularly between adipose and liver tissue, represents another prominent role of EVs in NAFLD. Adipocyte-derived EVs have been shown to exacerbate insulin resistance and impair gluconeogenesis in liver tissue, promote hepatic inflammation via monocyte chemoattractant protein-1 and IL-6 [66] and contribute to tissue remodelling via alteration of TIMP-1 expression in hepatocytes and HSCs [67]. Indeed, exposure to palmitate disrupts the circadian rhythm of lipid metabolism in both adipocytes and hepatocytes in vitro and in vivo murine tissue [68,69]. Recently, however, Zhao [70] showed that the liver facilitates early responses to lipid overload. Increased hepatic EV release targeted to adipocytes contained miR-122 and other miRNAs driving lipid deposition. While several studies have reported alterations in EV numbers and miRNA cargo in vitro and in human NAFLD sera [52,53,62], the degree to which the global circulating pool reflects changes in specific tissues remains uncertain, and so, important changes in liver EVs may be diluted. This is apparent in a study of adipocyte-specific knockout of Dicer—an enzyme crucial to miRNA maturation and sorting. More than 80% of exosomal miRNAs were significantly decreased despite invariable exosome numbers; suggesting that adipose tissue is a major contributor to total exosomal miRNA in circulation [71]. Hence, further investigation into the contributions of different tissue types in NAFLD compared to healthy serum is needed to properly contextualise changes in EVs.

## 7. Use of Small EVs as Diagnostic Biomarkers in NAFLD

Small EVs provide an attractive means of diagnosing disease, as their accessibility in blood and many other biological fluids enables their use as a “liquid biopsy” [7,8]. Just as they communicate across cells within and between different organs in the body, EVs may be harnessed to communicate to clinicians important molecular information that provides a snapshot of organ function. EVs have a longer half-life in circulation than proteins and nucleic acids and protect this encapsulated cargo [29]. Characteristics of circulating EVs, including abundance and molecular composition, have been shown to change significantly in disease states and this would be particularly useful to understand in conditions such as NAFLD, where current methods of diagnosis could be improved with respect to accuracy and invasiveness [2].

### 7.1. Circulating miRNAs as Biomarkers in Human NAFLD

miRNAs circulate in the blood contained in EVs or bound to argonaute proteins (mostly Ago2). They are attractive as biomarkers due to their stability, ease of detection and tissue-specific expression [72,73]. The majority of studies on the use of miRNA as biomarkers for NAFLD are based on detection directly from serum and, while a long list of miRNAs have been postulated, the most commonly reported include miR-122, -192, -34a, -16 and -21 [74]. The significance of specifically isolating the circulating EV-derived source of biomarkers from a crude ‘soluble pool’ has been highlighted in recent studies, most notable in in medical oncology, where the prognostic value of specifically quantifying EV-derived PD-L1 expression has been demonstrated [75,76]. It is important to consider the current evidence for miRNA markers, which is discussed here in this context, and to recognise that this field may be reshaped by studies that utilize EV isolating techniques prior to quantification of miRNA markers.

Several studies report significant alterations in miRNA profile in disease and the potential of individual or combined expression to outperform classical non-invasive diagnostic biomarkers. Table 1 summarises these reports, including AUROC values for diagnosis of SS, NASH or fibrosis. Early work by Cermelli [77] found serum levels of miR-122, -16 and -34a increased in NAFLD patients compared to healthy controls by 7.2-fold, 5.5-fold and from undetectable levels up to 2.2 × 10^4^ copies/mL, respectively. Division of the disease cohort by NAS revealed further increases (2–3-fold) in miR-122 and -34a from SS to NASH, while miR-16 levels were similar. miR-122 and -34a also positively correlated with fibrosis stage and inflammatory activity. Better performance in predicting early disease stage was observed for miR-122 and miR-16 compared to ALT, while miR-34a and -122 differentiated SS and NASH with moderate accuracy. Another study involving ultrasonically-assessed NAFLD found significantly increased serum miR-122 and -34a, as well as increased miR-451 and -21 in males. Only miR-122 correlated with severity of steatosis in both sexes [78]. Miyaaki [79] also showed increased miR-122 in the serum of severe steatosis patients compared to mild and healthy controls, with good accuracy (AUROC 0.82) in NAFLD diagnosis. Becker [26] reported similar AUROC values using a sum scoring system for serum miR-122, -192 and -21 in NASH. Significant positive correlations were found between miR-122 and -192 with CK-18, and the addition of CK-18 levels improved diagnostic performance marginally.

Based on a global serum profiling approach, Pirola [80] found most notable upregulation of miR-122, -192 and -375 in NASH and SS serum and associations with disease severity. miR-122 could also distinguish fibrosis. However, the miRNAs demonstrated only moderate accuracy in discriminating mild and severe disease and performed no better than liver transaminases. Another study by Salvoza [81], however, found significantly elevated serum miR-122 and -34a, but no differences in miR-375 and -21 in NAFLD and no significant correlations with histological features. miR-122 was only slightly better than ALT in diagnostic performance, but was again more sensitive in earlier stages than the liver enzyme. Work by Liu [27] found increased serum expression of miR-122, -192 and -34a with fold changes of 7.9, 4.0 and 2.8, respectively, and these correlated with steatosis and inflammatory activity. miR-34a was superior to multiple clinical biomarkers (ALT, CK-18, FIB-4 score and AST to platelet ratio index, APRI) in diagnosing NAFLD (AUROC 0.811). miR-16 was also upregulated in disease and correlated with fibrosis. However, its predictive value was less than that of FIB-4 and APRI. Further, whilst miR-34a could distinguish NAFLD from chronic hepatitis B patients, serum miR-122 and -192 levels were greater in the latter cohort, thereby limiting their use as NAFLD-specific markers [27].

Despite large fold increases in miR-122 observed in previous studies [27,77], a Finnish population-based study with a considerably larger sample size found an expression fold change of 1.55 in patients with ultrasonically-assessed NAFLD. Further, the accuracy of miR-122 was comparable to liver enzymes in predicting all fatty liver and the authors doubted the clinical significance of this biomarker in the general population [82]. Importantly, several other studies noted significant positive correlations of miRNAs, including miR-122, -192 and -34a, with liver enzymes [26,77,79,83]. While in these reports, the miRNAs generally performed slightly better than ALT or AST in predicting NAFLD, this might explain their currently limited value as clinical markers. Just one study did not detect differences in serum miR-122 and -34a, rather reported significantly decreased miR-181d, -99a, -197 and -146b in NASH serum with moderate to good accuracy in discriminating disease [84]. Finally, a study by Tan [72], involving RNA deep sequencing in NAFLD and control sera, followed by RT-qPCR validation, identified a panel of miRNAs (-122, -1290, -192 and 27b) that together exhibited good diagnostic accuracy that was not affected by adjusting for NAS. The emerging notion that combinations of miRNAs may improve diagnostic performance was also supported by López-Riera [74]. This recent work validated much of the previously postulated miRNA dysregulation in NAFLD and found that serum ratios of miR-34a/197, miR-192/30c and miR-27b/30c could discriminate SS and NASH, and that miR-27b, -192, -22, -30c and miR-197 were NAFLD-specific compared to a cohort of drug-induced liver injury patients. The expression pattern of serum miR-122 in the development of fibrosis in NAFLD is contentious within the literature, as both positive [77,80] and inverse [79] correlations have been described. More recently though, Akuta [83] suggested biphasic expression, with early increases in serum levels, followed by a decrease in advanced fibrosis. A strength of this work was the use of serial biopsies which supported earlier findings by the same group that miR-122 increased in serum through fibrosis stages 0-3 and decreased in stage 4 [85]. The reported increases in serum miR-122 are consistently correlated with decreased hepatic expression [77,79,80,86] and linked to the development of HCC [83].

### 7.2. EV-Encapsulated miRNAs

Results from the serum-based human studies described above are encouraging, but the diagnostic performance of these miRNAs is insufficient. Further, interesting changes in miRNA cargo of sEVs have been observed in in vitro models of lipotoxicity, yet data on differential expression profile in EVs from diseased patients are limited. EVs may potentially improve biomarker discovery and diagnostic accuracy, as it is argued that circulating miRNAs are concentrated in exosomes and that their isolation increases the sensitivity and reproducibility of miRNA analyses compared to directly from serum [87,88]. Further, we envision that new developments around the selective isolation of liver-derived EVs will offer the intriguing potential to identify a range of EV-based markers with increased specificity to NAFLD in the context of other chronic liver diseases. One study involving microarray analysis on EVs isolated by ExoQuick reported differential miRNA expression that could distinguish chronic hepatitis C, hepatitis B, NASH and healthy liver with 87.5% accuracy. It was acknowledged that this analysis likely included not only exosomal miRNA, but protein precipitates due to the use of ExoQuick [87]. Indeed, understanding the preferential distribution of circulating miRNAs in EVs or protein complexes may aid diagnostic applications. miR-122, in particular, is liver specific, but not disease specific, as its expression is altered in liver diseases of various aetiology such as HCC, viral hepatitis, cirrhosis and alcoholic or drug-induced liver injury [72]. Bala [89] reported that serum miR-122 was enriched in EVs in alcoholic and non-alcoholic inflammatory mouse models, but found mostly in the protein-rich fraction in acetaminophen-induced liver injury. Povero [53] also reported a shift from Ago2-bound miR-122 to EV encapsulation in a NASH mouse model. Pirola [80] demonstrated, via Ago2 complex immunoprecipitation, that only a small fraction of circulating miR-122 is Ago2 bound in human NAFLD serum, but did not identify EVs as the alternative source. While these reports add another layer of complexity to disease-associated changes in circulating miRNA expression, advancing knowledge in this area may further support the potential of EVs to diagnose NAFLD and discriminate steatohepatitis with greater sensitivity and specificity.

### 7.3. Current Challenges in EV-Based Methodology

sEVs comprise a population of vesicles that vary markedly in terms of diameter, density, membrane structure, surface protein expression and composition of encapsulated cargo. This heterogeneity exists not only due to differing cell types but also in vesicles released by the same cell [33]. The overlap in the size of sEV subtypes originating from distinct pathways of biogenesis necessitates the use of other defining features—of which, there remains a lack of understanding [7]. Consequently, there are major challenges in isolating specific EV subtypes for detailed investigation of their separate biological functions. Furthermore, a consensus has not been reached in regards to the optimal method for EV isolation from biological fluids and this lack of standardisation limits translational value [34]. Particular strategies, such as differential ultracentrifugation, density gradient separation, immunoprecipitation, and size-exclusion chromatography, are employed based on the aims of the study, type of downstream analyses, and impurities deemed acceptable [7,33,34]. Indeed, disparities in protein and RNA content are evident across EV purification methods [30]. Immunoprecipitation is an affinity-based method that produces relatively pure subtype populations via targeting specific surface proteins [33]. Examples of surface markers include CD81 for exosomes and Annexin A1 for MVs, but as the complex mechanisms of biogenesis and release are further elucidated and subtype definitions evolve, so too must the approach to their delineation [40]. Further complexity arises across in vivo studies, as the pool of EVs accessible in the blood reflect the contributions from multiple tissues. EV-derived markers, including miRNA, may be expressed in more than one tissue or exhibit differential expression in various diseases. As this has the potential to confound biomarker analyses, targeting tissue-specific surface markers in the process of EV isolation may be necessary for the sake of reproducibility and disease specificity [2,29,90]. While this field is still in its infancy, some useful markers have emerged in the context of liver-derived EV isolation, which include asialoglycoprotein receptor 1 (ASGR1) or cytochrome P450 2E1 (CYP2E1), and have the potential to advance in vivo work in NAFLD [2].

## 8. Conclusions

NAFLD is a highly prevalent and complex condition that is frequently observed in patients with a background of other systemic manifestations of metabolic disruption such as obesity and type 2 diabetes. While inherently benign in the early stages, approximately one in three patients progress to NASH, a more severe symptomatic form of the disease, which is associated with a marked increase in the development of cirrhosis and liver cancer. The lack of robust minimally invasive approaches to diagnose and track the disease, particularly in the early asymptomatic stages, represents a major ongoing challenge to both patient management and drug development for this disease. Of the various minimally invasive tests that have been explored, circulating EVs and their cargo present a highly promising capacity to reflect disease processes in the liver. However, there is still a range of important questions to address for the application of sEVs in NAFLD diagnosis to come to fruition. Of interest may be further elucidating the precise mechanisms of EV subtype biogenesis in the liver as well as the influence of toxic fatty acids on these pathways and packaging of cargo that may be quantified as biomarkers [7,35]. Further studies may also explore temporal changes and inherent variation in the characteristics of sEVs and cargo of interest, in healthy individuals and over the natural history of NAFLD, in order to gain insight into their capacity to establish disease presence, stage and individual risk of progression.

It is the view of these authors that the key determinant of the future role of EV-derived markers in the diagnosis and staging of NAFLD is based on the methodological advancement achieved by the selective analysis of EV-derived markers—specifically, the change in diagnostic performance resulting from analysis of circulating miRNA markers post-EV isolation, particularly when utilizing tissue (liver)-specific isolation techniques. Similarly, the capacity to quantify non-miRNA EV-derived NAFLD biomarkers, such as EV-derived Vanin-1 and PNPLA3 proteins, may provide useful complementary insights to distinguish NAFLD from other forms of liver damage.

## Figures and Tables

**Figure 1 jcm-09-02032-f001:**
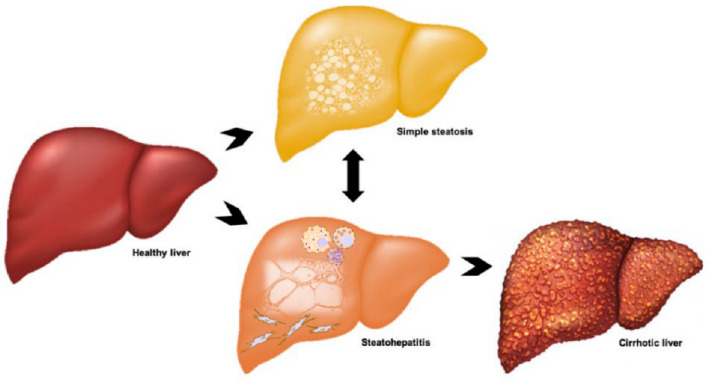
Clinical spectrum and progression of non-alcoholic fatty liver disease (NAFLD). Patients may present with simple steatosis due to lipid accumulation in hepatocytes, or non-alcoholic steatohepatitis (NASH), which is characterised by inflammation, hepatocyte death and fibrosis. Fibrosis can progress to cirrhosis and these patients have an increased risk of hepatocellular carcinoma.

**Figure 2 jcm-09-02032-f002:**
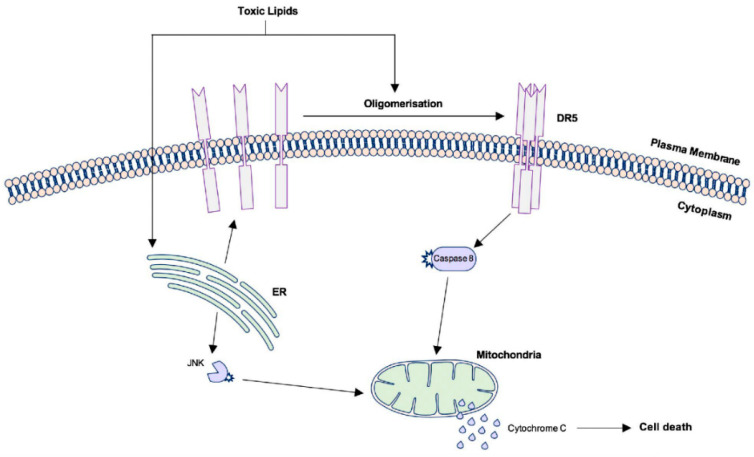
Signalling pathways leading to apoptotic cell death in lipotoxic hepatocytes. Palmitate and other toxic lipid metabolites trigger ligand-independent activation of TRAIL-R2 (DR5), leading to cell death via caspase 8 and egress of pro-apoptotic factors from the mitochondria. Toxic lipids also contribute to ER stress which may initiate the intrinsic pathway of apoptosis. ER: Endoplasmic reticulum; TRAIL-R2/DR5: TNF-related apoptosis-inducing ligand receptor 2/death receptor 5.

**Figure 3 jcm-09-02032-f003:**
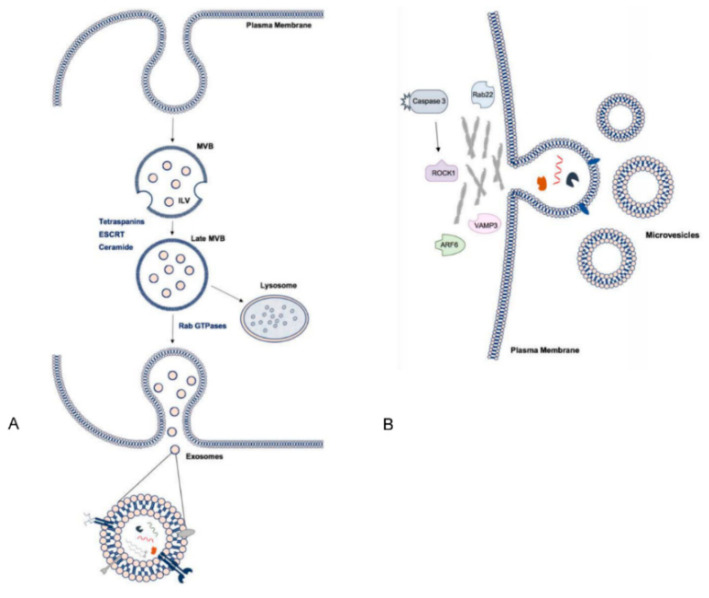
Pathways of small extracellular vesicle (sEV) biogenesis and summary of cargo. (**A**) Intraluminal vesicles (ILVs) are formed by invagination of the early endosomal membrane and subsequently secreted as exosomes. Multivesicular body (MVB) formation may be dependent on tetraspanins and ESCRT machinery or ceramides and trafficking to the membrane involves Rab GTPases. (**B**) Microvesicles (MVs) are shed directly from the plasma membrane. Various proteins are involved at the site of membrane blebbing and in processing cargo. EV cargo includes nucleic acids, proteins and lipids. **ESCRT:** endosomal sorting complex required for transport.

**Figure 4 jcm-09-02032-f004:**
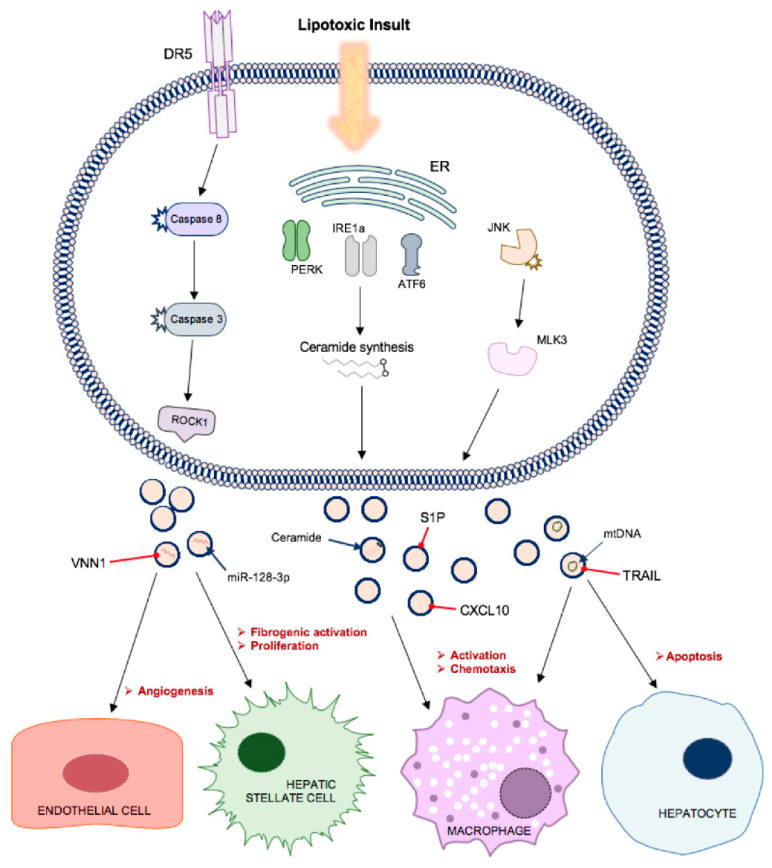
Lipotoxic hepatocytes release increased numbers of small EVs containing specific cargo that contribute to disease processes in non-alcoholic fatty liver disease. Multiple signalling pathways, including via ER stress sensors, c-Jun N-terminal kinase (JNK) and caspase activation, promote the release of EVs with cargo that promote endothelial cell migration and angiogenesis, transdifferentiation of hepatic stellate cells to a profibrogenic phenotype, macrophage activation and chemotaxis and apoptosis of neighbouring hepatocytes.

**Table 1 jcm-09-02032-t001:** Summary of serum miRNA biomarker studies in patients with non-alcoholic fatty liver disease (NAFLD).

Serum miRNA	Disease-Associated Change (↑/↓)	Sample Size	Verified against Tissue Biopsy (Y/N)	AUROC	Reference
**miR-122** **miR-16** **miR-34a**	↑	19 healthy, 18 NAFLD (NAS 1–4), and 16 NASH NAS (5–7)	Y	0.93 (SS) and 0.7 (NASH) 0.96 0.75	Cermelli, Ruggieri [77]
**miR-122, miR-34a** **miR-451, miR-21**	↑ (both sex) ↑ (males)	311 healthy, 73 mild NAFLD and 19 severe NAFLD	N	N/A	Yamada, Suzuki [78]
**miR-181d** **miR-99a** **miR-197** **miR-146b**	↓	20 healthy and 20 NAFLD	Y	0.86 0.76 0.77 0.75	Celikbilek, Baskol [84]
**miR-122**	↑	52 patients with mild (<33% steatosis) or severe NAFLD (>33%)	Y	0.82	Miyaaki, Ichikawa [79]
**Panel incl. miR-122, -1290, -27b, -192, -148a, -99a**	↑	190 healthy and 275 NAFLD	Y	0.856 (panel)	Tan, Ge [72]
**miR-122** **miR-192** **miR-21**	↑	61 healthy, 50 NAFL and 87 NASH	Y	0.81 (combined)	Becker, Rau [26]
**miR-375** **miR-122** **miR-192**	↑	16 healthy, 16 SS and 16 NASH	Y	0.72 0.67 0.69	Pirola, Fernández Gianotti [80]
**miR-122**	↑	36 NAFLD at different stages	Y	N/A	Akuta, Kawamura [83]
**miR-34a** **miR-16** **miR-192** **miR-122**	↑	37 healthy and 48 NAFLD	Y	0.811 0.716 (fibrosis) - -	Liu, Pan [27]
**miR-122** **miR-885**	↑	724 healthy and 147 NAFLD	N	0.709–0.810 0.603–0.632	Raitoharju, Seppälä [82]
**miR-34a** **miR-122**	↑	36 healthy and 28 NAFLD	Y	0.781 0.858	Salvoza, Klinzing [81]
**miR-34a, miR-27b, miR-122, miR-192, miR-22** **miR-30c, miR-16, miR-197**	↑ ↓	17 healthy, 25 NAFL and 50 NASH	Y	0.81 (miR-34a/197) 0.78 (miR-192/30c)	López-Riera, Conde [74]

AUROC: area under the receiver operating curve NAFL: non-alcoholic fatty liver; NASH: non-alcoholic steatohepatitis; NAS: NAFLD activity score.

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
