# Peer review of "Role of Extracellular Vesicles in the Pathophysiology, Diagnosis and Tracking of Non-Alcoholic Fatty Liver Disease"

_jcm, 2020, doi:10.3390/jcm9072032_

Round 1

Reviewer 1 Report

In this review Newman and colleagues summarize the currently available literature on the role of exosomes in NAFLD. The review provides a timely, comprehensive and overall well-written overview of both, the potential of ECVs as mechanistic mediators of the disease and as minimally invasive biomarkers. Some specific comments and suggestions to make the work even better can be found below:

  • While I appreciate that background and context for NAFLD/NASH and exosomes is provided, I feel that a total length of the introduction of 9 pages is way too extensive (and is in fact longer than the remainder of the paper). I would recommend to remove or drastically shorten sections dedicated to disease burden, biopsies or elastography. I feel this would strongly increase readability.
  • Some sections would benefit from more structure. For instance, it would benefit readability if the single paragraph on page 12 would be broken up into multiple paragraphs.
  • Other studies have shown that adipocytes seem to contribute >80% of all ECVs in human serum (e.g. PMID 28199304). How do the changes in exosome number observed in the cited NAFLD studies relate to this data? Could palmitate feeding affect serum exosome abundance and cargo via adipocytes rather than hepatocytes?
  • It would be appreciated if the authors would further discuss the specificity of exosomes and exosomal cargo as biomarkers for NAFLD. For instance, miR-122 appears to be a rather unspecific marker for liver injury that also associates with viral infections or drug-induced liver injury, without added value over conventional transaminase measurements.
  • Conclusions: Where do the authors see the field? Do they envision that exosomes might be constitute useful biomarkers for diagnosis and staging of NAFLD in the near future? Or will they be limited to being unspecific injury markers? A personal summary would be good.
  • Also, the authors could discuss the potential of exosomes for cargo delivery as therapeutic modality.

Reviewer 2 Report

The review of Newman et al is an interesting read. The authors present recent work on extracellular vesicles and miRNA in context of NASH and NAFLD. However, the manuscript is mixing endosomes and microvesicles and the presented miRNA parameters are not as clear as suggested by their title.

Concerns:

  1. The authors should clearly distinguish between endosomes and microvesicles in their paragraphs. Especially, as the formation of these two vesicles is so different, they should be treated as two different entities throughout the manuscript.
  2. miRNAs are presented as connected to vesicles yet most of the data stems from serum or plasmaanalysis. The connection is only loose and should not be overestimated in a review.
  3. The section of cytokeratin 18 should contain the work of the group of Rautou et al (e.g. Gastroenterology 2012, 143, 166–176.e6.).
  4. Information that mir122 correlates with liver enzymes is missing (e.g. Obes Surg. 2020 Feb;30(2):391-400.) which could explain a similar detection capacity of mir122 and liver enzymes for NAFLD/NASH.
  5. The authors do not mention markers of circulating hepatic vesicles that have been described in the literature besides cytokeratin-18. As the review focuses on liver a general information about liver derived vesicles that can be detected in the plasma should be given.

Side note: I found it rather confusing to have two authors mentioned when citing papers then the usual et al after the first author.

Round 2

Reviewer 2 Report

The authors made all changes required.